# Depth Scaling in Graph Neural Networks: Understanding the *Flat Curve* Behavior

**Diana Gomes**                                             *diana.gomes@imec.be*
*Interuniversity Microelectronics Centre (IMEC), Belgium*
*Vrije Universiteit Brussel (VUB), Belgium*

**Kyriakos Efthymiadis**                          *kyriakos.efthymiadis@vub.be*
*Vrije Universiteit Brussel (VUB), Belgium*

**Ann Nowe**                                                   *ann.nowe@vub.be*
*Vrije Universiteit Brussel (VUB), Belgium*

**Peter Vrancx**                                            *peter.vrancx@imec.be*
*Interuniversity Microelectronics Centre (IMEC), Belgium*
*Vrije Universiteit Brussel (VUB), Belgium*

**Reviewed on OpenReview:** *https://openreview.net/forum?id=fdyHzoGT8g*

## Abstract

Training deep Graph Neural Networks (GNNs) has proved to be a challenging task. A key goal of many new GNN architectures is to enable the depth scaling seen in other types of deep learning models. However, unlike deep learning methods in other domains, deep GNNs do not show significant performance boosts when compared to their shallow counterparts (resulting in a *flat curve* of performance over depth). In this work, we investigate some of the reasons why this goal of depth still eludes GNN researchers. We also question the effectiveness of current methods to train deep GNNs and show evidence of different types of pathological behavior in these networks. Our results suggest that current approaches hide the problems with deep GNNs rather than solve them, as current deep GNNs are only as discriminative as their respective shallow versions.

## 1 Introduction

Graph Neural Networks (GNNs) have been extensively used for graph representation learning in several different domains, such as social sciences, drug design, biology and particle physics. Since the proposition of Graph Convolutional Networks (GCNs) (Kipf & Welling, 2017), new network architectures have emerged aiming to improve expressivity (Veličković et al., 2018; Xu et al., 2018a), decrease computational cost (Hamilton et al., 2017), handle graph oversmoothing (Chen et al., 2020; Zhao & Akoglu, 2020) and overcome harmful node-level heterophily (Yan et al., 2022; Pei et al., 2020; Bodnar et al., 2022).

Deep GNNs show very different trends from other deep learning methods. Unlike conventional neural networks used in other domains (e.g. CNNs in computer vision), for which increasing the number of stacked layers is generally associated with greater expressive power and improved performance, the basic versions of GNNs (such as GCN (Kipf & Welling, 2017) or SGC (Wu et al., 2019)) do not benefit from depth; on the contrary, increasing depth leads to a significant drop of performance (see GCN in Figure 1). This trend is generally explained by the oversmoothing problem – a phenomenon in which all nodes in a graph become indistinguishable as a consequence of multiple aggregations of node representations through message-passing operations.

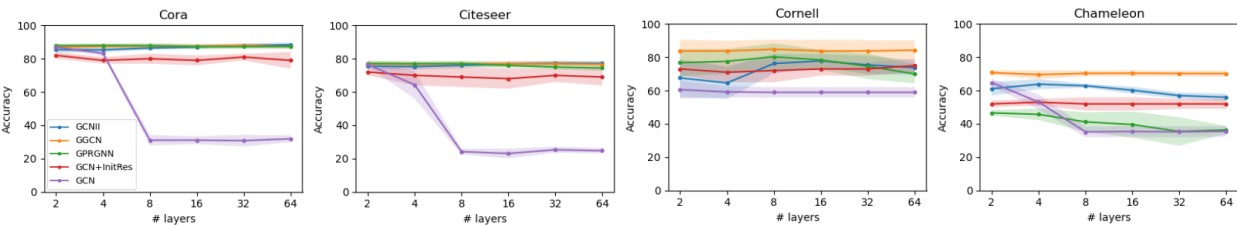

Figure 1: Examples of the *flat curve* behavior (performance stagnation over depth in GNNs). Results for GCNII, GGCN, GPRGNN and GCN were extracted from Yan et al. (2022). GCN+InitRes refers to results of this work (for $\alpha = 0.5$).

At the same time, the motivation to train deep GNNs still stands, as a $k$-layer GNN could potentially capture useful long-range dependencies by enabling the aggregation of relevant information $k$-hops away. This observation has led to the proposition of more complex network architectures, which do enable us to train deep networks to solve challenging tasks with adequate performance (e.g., GCN with initial residual (Gasteiger et al., 2018; Chen et al., 2020; Jaiswal et al., 2022), GCNII (Chen et al., 2020), GGCN (Yan et al., 2022), GPRGNN (Chien et al., 2021), $G^2$ (Rusch et al., 2023b)). However, these deep architectures often lead to equivalent or even deteriorated performance when compared to their shallow counterparts, as Figure 1 clearly shows. This opens unresolved questions regarding the pursuit of depth in GNNs.

Previous works have elaborated on the theoretical role of depth, but their conclusions often lack further empirical validation, and it is even possible to find contradictory arguments regarding the potential benefits of depth following theoretical reasoning (Keriven, 2022; Cong et al., 2021). Poor GNN performance due to apparent oversmoothing or loss of expressivity in deeper GNNs has also been investigated in some works (Yan et al., 2022; Li et al., 2018; Balcilar et al., 2020; Keriven, 2022; Oono & Suzuki, 2019), while other authors argued that it is not oversmoothing, but rather the training difficulty of GNNs that leads to poor results (Luan et al., 2023; Cong et al., 2021). This scenario evidences that the way that GNNs learn is still poorly understood, and dedicated empirical analyses on learning behaviour and depth scaling are in demand.

In this work, we aim to explain the plateau of performance over depth illustrated in Figure 1 by conducting an empirical study using eleven of the most common benchmark datasets for semi-supervised node classification tasks. We leverage GCN with initial residual (GCN+InitRes) as a case study, as these networks exhibit an analogous behavior to that of the remaining (more complex) architectures depicted in Figure 1 in terms of performance over depth (the *flat curve*). Our experiments show the extent to which this behavior manifests, and are followed by the study of the intermediate hidden node representations of several state-of-the-art deep networks to provide a justification for the observed phenomenon. These analyses disclose relevant empirical insights on how deep GNNs learn and whether/how they can actually leverage depth. We further discuss how these insights generalize between different deep architectures, and propose a depth evaluation protocol that should be adopted by other researchers to dismiss the possibility of pathological behavior in their networks. Our main contributions are:

- Bringing forward – for the first time, to the best of our knowledge – the discussion of how current methods are hiding the problems with deep GNNs rather than solving them, as these are only as discriminative as their shallow counterparts;

- Showing evidence of different types of pathological behavior in current deep GNN architectures, which manifest as redundant hidden embedding representations and seemingly useless/noisy layers;

- A protocol to identify pathological behavior when scaling GNNs with respect to depth (code publicly available[1]).

---

[1]https://github.com/dsg95/gnn-depth-eval

The manuscript is organized in five sections. Section 2 introduces some fundamental concepts behind deep GNNs (2.1), and proceeds to compare state-of-the-art approaches (2.2) and present a case study on the discriminative power of deep networks (2.3). Section 3 introduces previous methods for evaluating depth scaling in GNNs and other deep networks (3.1), proposes a depth evaluation protocol that overcomes some limitations of the previous (3.2), and discloses several types of inefficient learning behaviors in state-of-the-art deep GNNs following the application of our protocol (3.3). Section 4 discusses how our insights generalize between different deep GNNs and justify the *flat curve* phenomenon (4.1), including some additional reflections on how other well-known challenges of the field can be tied to the learning inefficiencies brought forward in this work (4.2 and 4.3); a discussion of limitations and future work wraps up this section (4.4). Section 5 summarizes the conclusions of the work.

## 2 Deep GNNs

### 2.1 Preliminaries

#### 2.1.1 Graph Convolutional Networks

We introduce GCNs as proposed by Kipf & Welling (2017) in the context of semi-supervised node classification. Considering a graph $G = (V, \mathbf{A})$, let us define $V$ as the set of all $n$ vertices $\{v_1, ..., v_n\}$ and $\mathbf{A}$ as an adjacency matrix of size $n \times n$ where $a_{ij}$ defines the edge between nodes $v_i$ and $v_j$: if $v_i$ and $v_j$ are connected, $a_{ij} \in ]0, 1]$; otherwise, $a_{ij} = 0$. Let us also define $\mathbf{D}$, the diagonal degree matrix of $G$, where the degree of $v_i$ can be defined as $d_i = \sum_j a_{ij}$. The normalized adjacency matrix of $G$ with added self-loops can now be written as $\hat{\tilde{\mathbf{A}}} = \tilde{\mathbf{D}}^{-1/2} \tilde{\mathbf{A}} \tilde{\mathbf{D}}^{-1/2}$, where $\tilde{\mathbf{A}} = \mathbf{A} + \mathbf{I}_n$ and $\tilde{\mathbf{D}} = \mathbf{D} + \mathbf{I}_n$. Each node $v_i$ is also associated with a $m$-dimensional feature vector $\mathbf{x}_i$. Stacking all $n$ feature vectors, we get a $n \times m$ feature matrix $\mathbf{X}$, which consists of the initial node representations $\mathbf{H}^0$.

$$\mathbf{H}^l = \sigma(\hat{\tilde{\mathbf{A}}} \mathbf{H}^{l-1} \mathbf{W}^l) \tag{1}$$

The output representations $\mathbf{H}^l$ of each graph convolution layer $l \in \{1, ..., L\}$ can now be defined by Equation 1, where $\mathbf{W}^l$ consists of a matrix of learned weights and $\sigma$ is a nonlinear activation function, such as ReLU. Finally, for node-level tasks, a classification head consisting of a linear layer and an activation function receives $\mathbf{H}^L$ and provides the final predictions $\mathbf{Y} \in \mathbb{R}^{n \times c}$, where $c$ is the number of classes.

#### 2.1.2 Oversmoothing

Given the smoothing nature of graph convolutions, training deep GNNs presents very specific challenges, in particular how to overcome graph oversmoothing. Oversmoothing is a consequence of the multiple aggregations of node representations in consecutive graph convolutional layers, which results in the loss of their expressive power as all nodes in the graph will exponentially converge to the same constant (Rusch et al., 2023a). Therefore, a network that produces oversmoothed node representations inevitably decreases its discriminative power to the point of random classification. As such, pursuing depth in GCNs inherently assumes that the network architecture accommodates a method that enables them to avoid this phenomenon.

### 2.2 Current Recipes for Depth Scaling

Inspired by the success of deep learning in other domains, searching for GNN layers that can scale with respect to depth has been a driver for researchers in the field, despite well-known bottlenecks (e.g., oversmoothing). Table 1 shows some examples of such layers. The following sections elaborate on the used methods and how these have been combined in layers that enable deep architectures. We further discuss what these approaches have in common and what may distinguish them.

### 2.2.1 Residual Connections

Residual connections are one of the most commonly employed methods to mitigate oversmoothing, thus enabling the training of deep GNNs. The terms *residual* and *skip* connections are often used interchangeably to refer to a family of methods that enable the flow of information directly from representations of previous layers to later, deeper ones in deep neural networks. If we stay closer to the initial proposition of residual connections (He et al., 2016), we can define them by Equation 2, as also proposed in the original GCN work (Kipf & Welling, 2017) with the purpose of containing oversmoothing and the inherent performance drop associated with increasing depth.

$$\mathbf{H}^l = \sigma(\hat{\hat{\mathbf{A}}}\mathbf{H}^{l-1}\mathbf{W}^l) + \mathbf{H}^{l-1} \tag{2}$$

However, in its simplest form, this method only seems to be moderately effective, as results show that oversmoothing tends to be delayed but not fully prevented and we should expect a performance drop as we stack more layers. As such, subsequent works (Chen et al., 2020; Gasteiger et al., 2018) proposed a direct connection to the initial node representation in the form of Equation 3, where $\alpha \in [0, 1]$ is a parameter that controls the strength of the residual connection.

$$\mathbf{H}^l = \sigma(((1-\alpha)\hat{\hat{\mathbf{A}}}\mathbf{H}^{l-1} + \alpha\mathbf{H}^0)\mathbf{W}^l) \tag{3}$$

Other connections (dense (Guo et al., 2019), jumping knowledge (Xu et al., 2018b)) have also been proposed, but these have hardly managed to prevent the performance drop trend (Jaiswal et al., 2022). The initial residual connection defined by Equation 3 (GCN+InitRes), on the other hand, has been consistently assisting the training of deep GNNs without significant performance loss, which has motivated their adoption (either as described or with subtle adaptations) in recent works (Jaiswal et al., 2022; Kulatilleke et al., 2022; Feng et al., 2021; Liu et al., 2021; Zhang et al., 2022).

### 2.2.2 Other Methods

Several other methods have been proposed with the purpose of mitigating/reducing oversmoothing, such as *normalization and regularization* techniques (Zhao & Akoglu, 2020; Rong et al., 2019). Others use more complex layers and have been broadly categorized as *architectural tweaks* (Jaiswal et al., 2022) or *changing GNN dynamics* (Rusch et al., 2023a). Methods such as GraphCON (Rusch et al., 2022), GRAFF (Di Giovanni et al., 2022), $G^2$ (Rusch et al., 2023b), GGCN (Yan et al., 2022), GPRGNN (Chien et al., 2021) fall under this category. These methods frequently combine several other strategies into the same expression, including residual connections which have been deemed necessary to avoid their oversmoothing by Rusch et al. (2023b).

### 2.2.3 Comparison of State-of-the-art Approaches

Table 1 shows examples of layers that have been used to create deep architectures while attaining adequate performance. Despite the different levels of complexity of these layers, one can observe that they share some key ingredients:

- A way to preserve the information of earlier node representations in latter, deeper ones, either by adding a residual connection to $\mathbf{H}^{l-1}$ (GGCN, $G^2$), an initial residual connection to $\mathbf{H}^0$ (GCN+InitRes, GCNII), or by weighting all intermediate representations in the final embedding (GPRGNN);

- Coefficients (learnable or not) that weight the graph convolution term(s), and, thus, the effect of the increasingly smoothed representations in the latter embeddings.

While the first point helps preventing oversmoothing by having a sharpening effect in the later, more smoothed node representations, the second has the potential to hold back the smoothing effect of the graph

Table 1: Examples of layers that have been empirically shown to enable deep architectures. All layers include some type of residual connection or another method that enables the preservation of previous node representations. For a full explanation of all terms and coefficients, please refer to 5 of the Appendix and/or the original publications.

| | Method | Layer |
|---|---|---|
| Deep GNNs | GCN+InitRes | $\mathbf{H}^l = \sigma\left(\left((1-\alpha)\hat{\hat{\mathbf{A}}}\mathbf{H}^{l-1} + \alpha\mathbf{H}^0\right)\mathbf{W}^l\right)$ |
| | GCNII | $\mathbf{H}^l = \sigma\left(\left((1-\alpha)\dot{\mathbf{A}}\mathbf{H}^{l-1} + \alpha\mathbf{H}^0\right)((1-\beta)\mathbf{I} + \beta\mathbf{W}^l)\right)$ |
| | GPRGNN | $\mathbf{H}^L = \sum_{l=0}^{L} \gamma_l \dot{\mathbf{A}}\mathbf{H}^{l-1}$ |
| | GGCN | $\mathbf{H}^l = \mathbf{H}^{l-1} + \eta\left(\sigma\left(\alpha^l(\beta_0^l\hat{\mathbf{H}}^{l-1} + \beta_1^l(\mathbf{S}_{pos}^l \odot \dot{\mathbf{A}} \odot \mathbf{T}^l)\hat{\mathbf{H}}^{l-1} + \beta_2^l(\mathbf{S}_{neg}^l \odot \dot{\mathbf{A}} \odot \mathbf{T}^l)\hat{\mathbf{H}}^{l-1}\right)\right)$ |
| | G$^2$ | $\mathbf{H}^l = (1-\tau^l) \odot \mathbf{H}^{l-1} + \tau^l \odot \sigma\left(\dot{\mathbf{A}}\mathbf{H}^{l-1}\mathbf{W}^l\right)$ |

convolutions and decrease the speed of convergence to a constant value that defines oversmoothing. Continuous GNN approaches, such as GraphCON (Rusch et al., 2022) and GRAFF (Di Giovanni et al., 2022), were left out of the scope of this analysis for conciseness, but similar extrapolations could be performed for these architectures, which notably also include a term for preserving earlier node representations and smoothing control coefficients (see $\Delta t$ for GraphCON and $\tau$ for GRAFF).

Comparing these layers suggests that the occasional offset in accuracy between different models (Figure 1) is due to the unique terms/learnable coefficients of their layers, which can make them more or less expressive in some cases. However, the performance stagnation over depth phenomenon can still be verified (Figure 1), for the simplest of the models and the most complex ones alike, despite the architectural tweaks which add to their complexity.

## 2.3 On the discriminative power of deep GNNs: a case study for GCN+InitRes

The works presented in Table 1 show that we can design deep GNN architectures; but are there actual empirical gains in going deeper? And can we measure them? In this subsection, we conduct a series of analyses using the simplest of deep GNN models (GCN+InitRes) to provide further and more thorough insights on the potential of depth as a means to increase the discriminative power of GNNs.

GCN+InitRes layers can be defined by a graph smoothing term, equivalent to that of vanilla-GCNs, and an initial feature encoding term (the residual connection) with a sharpening effect. The influence of these terms in the layer-wise node representations is weighted by $\alpha$ and its symmetric (Equation 3), i.e., for $\alpha = 0$ we have the vanilla-GCN and for $\alpha = 0.5$ the smoothed representations have the same weight as the initial ones. We will expand on the problem of performance stagnation over depth for GNNs by exploiting GCN+InitRes properties, as the simplest of the deep GNN examples. We aim to find the practical benefits of scaling these models with respect to depth, while enhancing these empirical insights through comparisons with the base models that compose GCN+InitRes (i.e., GCN and MLP).

**Methodology.** We investigate the effect of using initial residual connections to train deep GCNs by conducting a thorough ablation study of the depth ($L$) and residual connection strength ($\alpha$). To that end, we conceive a framework, taking $L$ and $\alpha$ as input variables, where each layer assumes the form of Equation 3. We also consider a shallow vanilla-GCN and a 2-layer MLP as baselines for performance comparison. We use eleven of the most common dataset benchmarks, including homophilic and heterophilic networks (more benchmarking and training details in Appendix A). The results of the full ablation study are shown in Appendix B.1; Table 2 shows a compact version with the results for the best $\alpha$ for each network depth (further details on the comparison of models can be found in Appendix B.1.2).

**Benchmark patterns.** Comparing GCN+InitRes with the baseline models in Table 2, we can identify three distinct patterns within our benchmark datasets. *Case 1* groups benchmarks for which GCN+InitRes shows no evidence of superiority when compared to a vanilla-GCN. For these cases, a vanilla-GCN is able to deliver an equivalent or superior outcome compared to that of a GCN+InitRes, regardless of its depth. As such,

Table 2: Node classification accuracy of GCN+InitRes of increasing depth ($L$) and Vanilla-GCN and MLP baselines for eleven dataset benchmarks. **Bold** values correspond to largest mean accuracy (top performance) for each benchmark; Underlined values highlight results that show no statistically significant difference from the best performing GCN+InitRes on each benchmark (row-wise). [*Case 1*] Vanilla-GCN performance is superior or equivalent to GCN+InitRes. [*Case 2*] GCN+InitRes performs better than both baselines. [*Case 3*] MLP performance is superior or equivalent to GCN+InitRes.

| | Baselines | | GCN+InitRes | | | | | | | |
| | **Vanilla-GCN** | **MLP** | $L=1$ | $L=2$ | $L=4$ | $L=8$ | $L=16$ | $L=32$ | $L=64$ | $L=128$ |
|---|---|---|---|---|---|---|---|---|---|---|
| *Case 1* | | | | | | | | | | |
| Citeseer | .73±.02 | .68±.06 | **.74±.02** | .72±.02 | .70±.06 | .69±.06 | .68±.06 | .70±.04 | .69±.05 | .71±.03 |
| Squirrel | **.44±.01** | .33±.02 | .39±.02 | .41±.02 | .41±.02 | .42±.02 | .42±.02 | .42±.01 | .42±.01 | .42±.02 |
| Chameleon | **.60±.03** | .50±.03 | .49±.03 | .55±.03 | .54±.03 | .54±.03 | .53±.03 | .53±.03 | .54±.02 | .54±.03 |
| Cora | .82±.03 | .68±.05 | **.84±.02** | .82±.01 | .81±.02 | .80±.03 | .80±.03 | .81±.02 | .81±.02 | .81±.03 |
| Physics | **.97±.00** | .95±.03 | **.97±.00** | **.97±.00** | **.97±.00** | **.97±.00** | **.97±.00** | **.97±.00** | **.97±.00** | **.97±.00** |
| *Case 2* | | | | | | | | | | |
| Pubmed | .87±.01 | .86±.02 | **.89±.01** | **.89±.01** | **.89±.01** | **.89±.01** | **.89±.01** | **.89±.01** | **.89±.01** | **.89±.01** |
| CS | .93±.00 | .91±.03 | **.95±.00** | **.95±.00** | **.95±.00** | **.95±.00** | **.95±.00** | **.95±.01** | **.95±.01** | **.95±.00** |
| *Case 3* | | | | | | | | | | |
| Cornell | .43±.06 | **.75±.02** | .42±.05 | .73±.04 | .73±.06 | .74±.05 | .73±.04 | **.75±.04** | .75±.04 | .74±.05 |
| Texas | .59±.07 | **.82±.04** | .51±.13 | .78±.05 | .81±.04 | .80±.06 | .81±.05 | .80±.03 | .80±.06 | .80±.05 |
| Wisconsin | .51±.08 | **.87±.02** | .59±.05 | .85±.05 | .86±.03 | .84±.03 | .84±.04 | .84±.03 | .84±.05 | .84±.03 |
| Actor | .27±.01 | .35±.04 | .34±.01 | .36±.01 | **.36±.01** | .35±.01 | .35±.01 | .35±.01 | .35±.01 | .35±.01 |

adding a sharpening component in the form of an initial residual connection seems not to bring an evident empirical benefit in these cases. *Case 2* evidences benchmarks for which GCN+InitRes is consistently superior than its basic components alone. For these benchmarks, the combination of a smoothing and sharpening component in a single model is able to marginally improve the results, thus evidencing a benefit of practical utility. This benefit, however, seems to be independent of depth, as both shallow and deep GCN+InitRes models were able to deliver top performance of classification. Concerning *Case 3* benchmarks, we verify that GCN+InitRes achieves top performance for all $L \geq 2$, showing equivalence to the performance of a feature-only MLP in this case.

**Deep GCN+InitRes are as discriminative as their shallow counterparts.** Our results support that, while GCN+InitRes can be more discriminative than the base models that compose it (*Case 2*), using this method as a means to pursue depth shows no empirical benefit in any scenario. In fact, we find that there is always at least one shallow GCN+InitRes network (up to 2 layers) able to reach top performance.

## 3 Depth Evaluation

Benchmark results such as the ones disclosed in our case study (2.3) and Figure 1 make us question how we have been evaluating depth for GNNs. Works that conduct analyses on deep versions of new GNNs have been mostly relying on measuring whether they can successfully avoid oversmoothing, but have hardly evaluated whether progressively stacking more layers actually increases the discriminative power of the network. Depth scaling claims have been frequently supported by insufficient benchmarking tables including a single performance metric, frequently accuracy (mean and standard deviation), and consider a method to be superior if it exhibits the highest mean absolute accuracy, even if they have high, overlapping standard deviations and marginal performance gains (Chen et al., 2020; Bodnar et al., 2022; Rusch et al., 2023b).

This section introduces some background on previous depth evaluation approaches and where they fail. We further propose a new protocol that addresses some of their challenges, based on which we disclose a set of learning inefficiencies in state-of-the-art deep GNNs.

### 3.1 Background

#### 3.1.1 Insufficiency of Oversmoothing Measures

Previous works have evaluated deep GNNs' capacity to overcome oversmoothing by studying the Dirichlet energy (Jaiswal et al., 2022; Rusch et al., 2023a). Dirichlet energy can be given by Equation 4 and can essentially be described as a metric of local node similarity which satisfies useful conditions, namely if $\mathbf{X}_i$ is constant for all nodes $v_i \in V$, $\mathbb{E}(\mathbf{X}) = 0$. As such, the layer-wise exponential convergence of $\mathbb{E}(\mathbf{X})$ to 0 defines oversmoothing with respect to this measure.

$$\mathbb{E}(\mathbf{X}) = \sqrt{\frac{1}{n} \sum_{i \in V} \sum_{j \in N_i} ||\mathbf{X}_i - \mathbf{X}_j||_2^2} \tag{4}$$

Nonetheless, to evaluate the need for depth, it is not sufficient to just evaluate whether we are avoiding oversmoothing. In fact, according to Rusch et al. (2023a), the simple addition of a bias in each layer is enough to keep the Dirichlet energy from exponentially dropping to zero as we increase depth; however, this does not directly imply that we are gaining expressivity and deriving more meaningful node representations by stacking more layers.

#### 3.1.2 Finding Pathological Behavior in Deep Networks

Central Kernel Alignment (CKA) was introduced by Kornblith et al. (2019) as a robust measure of the relationship between feature representations within the same network and across different networks. In particular, the authors show that CKA (Equation 5) is invariant to orthogonal transform and isotropic scaling and can be used to understand network architectures, revealing when depth can become pathological in neural networks representations (e.g., when a big part of the network produces representations that are very similar and, thus, redundant). Following its introduction, CKA has been extensively used to study how deep neural networks learn representations (Nguyen et al., 2020; Raghu et al., 2021), although never to understand the specific case of deep GNNs.

$$Linear\ CKA(\mathbf{H}^l, \mathbf{H}^k) = \frac{||\mathbf{H}^{k^T}\mathbf{H}^l||_F^2}{||\mathbf{H}^{l^T}\mathbf{H}^l||_F \cdot ||\mathbf{H}^{k^T}\mathbf{H}^k||_F} \tag{5}$$

Recently, some researchers have been pointing out some handicaps of CKA and defending that its interpretation can be counterintuitive (Davari et al., 2022). Thus, it can be prudent to complement it with other methods that were previously employed to measure representation similarity, such as linear regression. Such methods have been used to validate CKA in its original work, and we argue that they can lead to complementary information towards understanding how deep networks are learning.

### 3.2 Depth Evaluation Protocol for GNNs

In order to study the representations learned by GNNs over depth, we propose a protocol that inspects the layer-wise hidden node representations with two different measures of similarity—Logistic Regression and linear CKA:

1. Train a Logistic Regression classifier on the intermediate hidden representations $\mathbf{H}^l \in \{1, ..., L\}$ of the same network, as a metric of the linear separability of the clusters at each consecutive layer;

2. Compute CKA to measure the pairwise similarity of hidden node representations of layers $l, k \in \{1, ..., L\}$ within the same GNN.

Ideally, hidden node representations of consecutive layers should lead to increasingly more separable clusters (evidenced by higher accuracies if classified by Logistic Regression models) and progressively become more

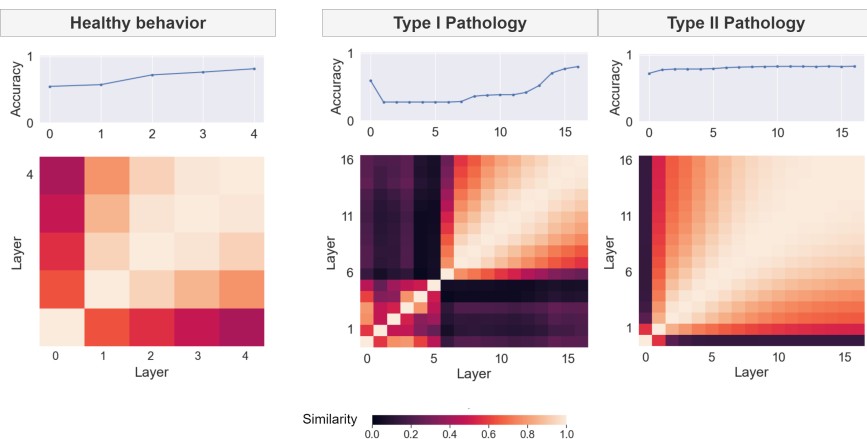

Figure 2: Examples of healthy vs. pathological learning behaviors. Layer-wise Logistic Regression is measured in terms of accuracy of classification using the intermediate hidden representations $\mathbf{H}^l \in \{1, ..., L\}$; pairwise CKA of layers $l, k \in \{1, ..., L\}$ is represented by the similarity matrices. Type I Pathology manifests as a sequence of noisy layers (particularly evident for $l < 6$); Type II Pathology manifests as a sequence of nearly identical layers (particularly evident for $l > 10$).

dissimilar from the earlier node representations of more shallow layers, as stacking more layers is expected to increase the discriminative power of the network. For concision, we will call this a *healthy* behavior (Figure 2); a deviation from this trend will be referred to as a *pathology*—a term already coined by previous works (Kornblith et al., 2019) for other types of deep learning models.

Kornblith et al. (2019) used Logistic Regression classification as a reference to validate CKA's efficacy in detecting depth-related pathologies. In this case, we propose an analogous protocol, in which analysing the linear separability of the clusters over depth is not irrelevant/redundant, but complementary. That is because, not only are we interested in identifying cases in which hidden embeddings might show high similarity (and thus be deemed redundant), but also whether we are progressively increasing the discriminative power of the network by stacking more layers. This latter analysis cannot be fully given by CKA alone (e.g. if $\mathbf{H}^l$ and $\mathbf{H}^{l+1}$ evidence low similarity, we cannot know if this dissimilarity is beneficial or not without the assistance of a complementary method able to show whether the classes are becoming progressively more separable).

### 3.3   Pathological Depth in GNNs

We apply our depth evaluation protocol to three deep GNN models: GCN+InitRes, GCNII, $\mathrm{G}^2$. Our experiments intend to verify the hypothesis of pathologic behavior in deep GNNs as a justification for the *flat curve* phenomenon. We show the results for one benchmark of each case surveyed in Table 2, to remain succinct while showing evidence of different patterns of behaviors.

**Methodology.** We evaluate depth through layer-wise inspection of hidden node embeddings, according to the protocol proposed in the previous section. Higher accuracy in Logistic Regression classification means that the clusters are more linearly separable. Representations with low similarity should lead to a CKA close to 0 and highly similar representations to a CKA of up to 1. Figure 3 illustrates the results of this protocol for three different deep GNN models (16/64-layers) on three benchmark datasets.

#### 3.3.1   Identification and Categorization of Pathologies

Our depth evaluation protocol shows evidence of unhealthy behavior for all of the studied deep GNNs. We categorise such behavior as two different types of pathologies. Figure 2 shows an example of healthy learning behavior as opposed to the pathological trends identified in this work. The following paragraphs elaborate on how these pathologies manifest in the different deep GNNs (Figure 3).

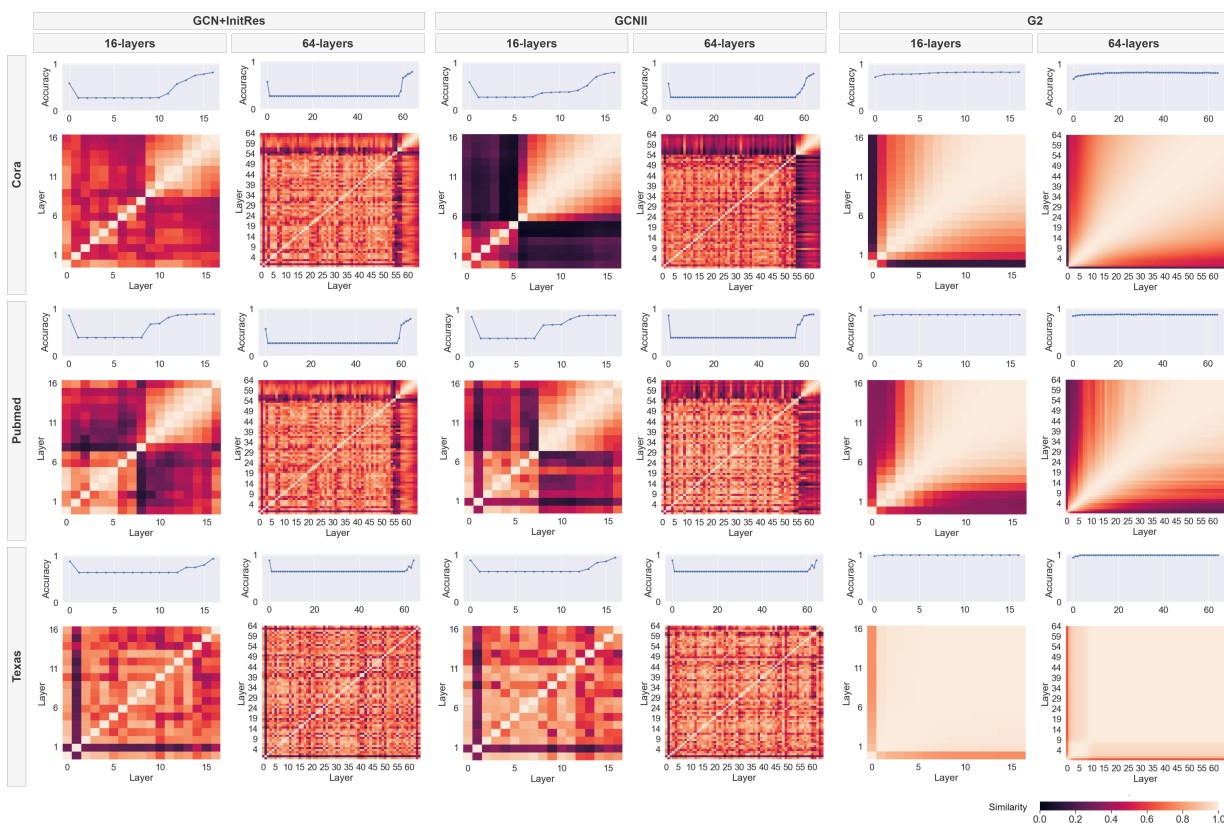

Figure 3: Results of our depth evaluation protocol for GCN+InitRes, GCNII and $G^2$ (16/64-layers) on three benchmarks (Cora, Pubmed, Texas). Layer-wise Logistic Regression is measured in terms of accuracy of classification using the intermediate hidden representations $\mathbf{H}^l \in \{1, ..., L\}$; pairwise CKA of layers $l, k \in \{1, ..., L\}$ is represented by the similarity matrices.

**Type I Pathology.** One of the major inefficiencies that we can observe in the examples of Figure 3 is a succession of noisy layers. This manifests as a sequence of layers that show no specific pattern in terms of similarity (visually producing a noisy square in the similarity matrix) and whose hidden representations prove not to be linearly separable (Logistic Regression accuracy equivalent to random classification). We call this pattern a *Type I pathology*. This pathology is very evident in the early layers of GCN+InitRes and GCNII networks, and gets more and more pronounced as we increase network depth because the number of noisy layers seems to increase proportionally with the overall depth of the network. Texas seems to be the benchmark that is more affected by this pathology (only the last 3-4 layers appear to contribute to improve the discriminative power of the network, regardless of its overall depth).

**Type II Pathology.** The last layers of GCN+InitRes and GCNII models for Cora and Pubmed seem to be able to progressively learn, but we can also verify an occasional redundancy of some consecutive layers. This redundancy is showed by semi-constant accuracy when it comes to the linear separability of the clusters and nearly identical node representations (similarity close to 1). In practice, this means that these layers could potentially be pruned with little expected consequences for the final performance of the network. We call this a *Type II pathology*. The impact of this pathology in the overall outcome is very limited for the example described above (as we only observe it for a small number of consecutive layers), but the same pathology can be easily identified for large parts of the $G^2$ networks. The extent to which we observe this pathology in $G^2$ models is particularly cumbersome for Texas, for which the large majority of every network shows identical node representations.

**Severe cases.** The severity of these pathologies varies with benchmark, model type and depth, but the phenomenons are the same. Both types of pathologies can coexist in the same network, with different levels

of severity (e.g., 16-layer GCN+InitRes on Pubmed). The fact that Texas exhibited the most exacerbated examples of the described pathologies is very significant, as Texas is a benchmark for which a feature-only MLP remains relevant by delivering state-of-the-art results, even when compared with complex GNN architectures (see Case 3 of Table 2). This observation is not a coincidence, and similar patterns that show severe pathological behaviors can be found for the remaining Case 3 benchmarks (see examples in Figure 7 of the Appendix). This finding hints at the conclusion that the key ingredients that have been enabling GNNs to go deep also give them the ability to completely obliterate the usage of graph structure when these networks are not able to encode such structure productively.

### 3.3.2   Causes

Going back to the comparison of deep GNN layers in section 2.2.3, we can yet find that these inefficient learning behaviors can be a direct and logical consequence of the key ingredients that have been enabling GNNs to go deep.

Looking into GCN+InitRes and GCNII in Figure 3, we see similar patterns of unhealthy learning, mostly through the development of Type I pathologies. These networks are also the ones that use direct connections to the initial node embeddings $\mathbf{H}^0$ in their layers (Table 1). We hypothesise that such connections are responsible for enabling Type I pathologies, as the networks are able to recreate the information lost due to the excessive smoothing of the consecutive graph convolutions in later embeddings, by using the initial representations. For this reason, the first $p$ layers are essentially useless/noisy (see drop in accuracy to random classification level from $\mathbf{H}^0$ to $\mathbf{H}^1$ for all deep GCN+InitRes and GCNII experiments), while the last $L - p$ layers seem to be able to progressively learn useful embeddings. By learning to obliterate the usage of its first layers, the networks are able to deliver $\mathbf{H}^L$ with optimal smoothing for the downstream task. The same optimal level of smoothing could, however, be achieved with a shallower network.

Pathologies of Type II seem to be related with the coefficients that weight the graph convolution term and the residual connection. All studied models (GCN+InitRes, GCNII, $\mathrm{G}^2$) consider these coefficients to be symmetrical. The main difference between these approaches is that $\mathrm{G}^2$ learns a scalar per layer ($\tau^l$), while GCN+InitRes and GCNII consider a tunable hyperparameter ($\alpha$) that is fixed for all layers of the network. The results of Figure 3 show that $\mathrm{G}^2$ is particularly affected by Type II pathologies, especially in the final layers of the networks. The high similarity of consecutive node embeddings for the deeper representations suggests that, as depth increases, $\tau^l$ tends to zero, at which point $\mathbf{H}^l = \mathbf{H}^{l-1}$ and layers become redundant. For Texas, a dataset for which graph convolutions do not seem to be particularly useful, this can even mean repetitive representations for all $l > 1$. We hypothesize that for GCN+InitRes and GCNII networks, $\alpha$ is also playing a role in refraining the smoothing speed, as occasional repetitive representations can be found for the last $L - p$ layers that progressively learn (e.g., semi-constant accuracy and high similarity of 16-layer GCNII on Cora for $8 < l < 11$). Further analyses on the role of the residual connection strength ($\alpha$) in GNNs can be found in Appendix B.1.

## 4   Generalization of Insights Between Different Deep GNNs

This section aims to further bridge the conclusions of sections 2 and 3 by discussing two key points of our work:

1. Pathologies are a consequence of the addition of residual connections and the smoothing refraining coefficients to GNN layers (as discussed in 3.3.2);

2. All surveyed methods in Table 1 (i.e., works that claim depth scaling) rely on both of these "ingredients" to create deep architectures (as discussed in 2.2.3).

The direct correspondence between the causes of the pathologies and the identified key ingredients of depth scaling suggests that similar behavior should be expected whenever these components are part of deep GNN

architectures, i.e., our conclusions are not limited to the examples we inspected. For this reason, we expand on the generic relation between the "flat curve" and the pathologies (4.1), on the role of residual connections – one of the most frequently adopted and unquestioned methods in the literature – in GNN learning (4.2), and on how pathologies can not only justify the "flat curve" but also other puzzling phenomena whose causes still lack general agreement in the community (like the equivalence between GNNs and MLPs on some heterophilic benchmarks).

## 4.1  Performance stagnation over depth: the *flat curve*

The inefficient learning behaviours identified as pathologies in the previous section provide a deeper understanding of the results presented in Table 2 for GCN+InitRes and, consequently, the *flat curve* phenomenon.

**GCN+InitRes case study.** Case 1 benchmarks correspond to datasets for which the underlying structure of the graph encodes relevant information under the smoothing assumptions of graph convolutions. These benchmarks benefit from at least one graph convolution operation, which is why vanilla-GCN can perform on par with GCN+InitRes. In these cases, GCN+InitRes develops a pathology of Type I—which ultimately turns a large part of the networks into useless representations—and, occasionally, some cases of redundant representations (Type II); as such, these networks only turn out to effectively take advantage of a small number of convolutions, leading to results equivalent to those of a shallow vanilla-GCN. We hypothesise that this behavior is possible due to the initial residual connection, which enables direct flows of information from the first, non-smoothed levels into the deeper, potentially oversmoothed ones (as discussed in 3.3.2).

Analogously, for benchmarks of Case 2, we observe pathologies of both types. In this case, GCN+InitRes derives more meaningful representations than its baseline counterparts; however, a shallow GCN+InitRes of 1 or 2 layers is sufficient to achieve top performance. As such, stacking more layers leads to inefficient learning behaviors.

Finally, Case 3 benchmarks seem not to benefit from graph convolutions, as both a feature-only MLP and a GCN+InitRes can achieve top performance. Benchmarks of this case undergo the most severe consequences of the pathologies and are not able to leverage the large majority of the network. Developing Type I or Type II pathologies enables them to circumvent the smoothing effects of graph convolutions. This leads them to rely mostly on linear transformations of the initial features (only possible due to the initial residual connection), followed by non-linearities—hence the empirical equivalence to the MLP.

**Other deep GNNs.** These observations for GCN+InitRes could be generalized for the remaining, more complex deep GNNs, since they are sustained by the theoretical analysis of the layers' expressions that led us to conclude that they share some key ingredients. The fact that all of them combine terms that preserve previous, non-smoothed representations and have coefficients that can refrain the smoothing (see Section 2.2.3) strongly hints at the conclusion that these networks are undergoing learning handicaps analogous to the ones identified in this work as pathologies when they attempt to go deep. This would justify why we also verify the *flat curve* in their case, and is corroborated by the results exhibited in Figure 3 for GCNII and $G^2$. Showing exhaustive evidence of the pathologies for the remaining models is out of the scope of this work, but we encourage other researchers to do so. We also discuss the case of GPRGNN and GGCN in slightly more detail in Appendix B.2.

Since the pathologies are rooted in the addition of residual connections and smoothing controlling coefficients to each layer, and not in the convolution operation itself (which can be more or less expressive in some cases), we expect that our conclusions can extend to other architectures (see an example for GAT+InitRes in Figure 9 of Appendix B.3). Evidently, we do not advocate that the proposition of the new and more complex layers is useless. These models have proved their potential to deliver state-of-the-art results in several benchmarks, which validates their practical utility and relevance within the field of GNN research. We do, however, emphasize that, at this point, we could find no evidence of any benefit of practical utility when scaling them with respect to depth, both in the literature and in our own experiments. This finding makes us question whether the pursuit of depth within the field is being properly addressed, as current approaches seem to overshadow the problems with deep GNNs rather then solving them. For this reason, we encourage other researchers to evaluate their claims of appropriate depth scaling more thoroughly, and propose a protocol to assist them in doing so.

## 4.2 The role of residual connections

The importance of residual connections has been highlighted in previous works (Di Giovanni et al., 2022; Jaiswal et al., 2022). However, we clarify the extent of their contribution from an empirical perspective (explicitly for initial residuals, but with implicit assumptions for other architectures), by looking into the intermediate hidden node representations. By doing this, we bring forward, for the first time, the fact that top performing GNNs with residual connections can converge to a solution in which they avoid oversmoothing by reconstructing non-smoothed representations from earlier embeddings in later layers and disregard the noisy information of the previous ones (Type I pathology). This means that while residual connections can make GNNs more powerful to some level, by enabling the combination of smoothed with sharp representations, they are not effective as a means to achieve depth.

We verify this for GNN+InitRes, but we cannot rule out that analogous behavior might be observed for more complex architectures, as all the surveyed architectures include coefficients that enable them to preserve previous node representations for indefinite depth (Table 1). For this reason, we encourage other researchers to perform similar analyses as the ones performed in this work when proposing new layer types that enable deep architectures by pertaining residual connections (or analogous strategies of preservation of the information of early hidden representations), and not just to evaluate if oversmoothing is avoided using the Dirichlet energy (which can be kept constant without that actually meaning that GNNs are increasing their discriminative power, as discussed in 3.1.1).

## 4.3 Overcoming harmful heterophily

The methods that have been proposed to address oversmoothing have also often been empirically verified to help overcoming harmful heterophily. This has led to some works looking into the two concepts – oversmoothing and heterophily – at once (Yan et al., 2022; Di Giovanni et al., 2022; Bodnar et al., 2022), namely using the Dirichlet Energy. Our results also provide a justification for this phenomenon by suggesting that for cases in which we cannot leverage the smoothing component (because the graph structure is not helpful (Gomes et al., 2022)), networks that use earlier, non-smoothed embeddings can ultimately just resort to these for their final representations (by evidencing a Type I or Type II pathology). This implies that the same phenomena that causes inefficient depth scaling can also explain the equivalence to MLPs in these cases: if neighbourhood aggregation through graph convolutions does not lead to improved representations for the downstream task, networks will learn to resort to noisy or identity layers (i.e., in this case, optimal smoothing is found for $L = 0$). This conclusion is novel and justifies why these networks have been performing on par with MLPs for these cases, a problem that had been previously brought forward (Gomes et al., 2022; Luan et al., 2022), but never fully explained, to the best of the authors' knowledge.

## 4.4 Limitations and Future Work

While our protocol addresses the limitations of previous depth evaluation methods, it is mostly focused in assisting the identification and interpretation of learning inefficiencies and does not provide a final statistical metric that can be directly used to compare different methods. This can limit its potential application scenarios. However, we believe that it would not be possible to derive such insightful empirical analyses from a single metric alone, without incurring in limitations analogous to those of the methods that precede this work, due to GNNs' complexity. Our study also mainly addresses how different GNNs learn the most useful representations for a certain task. It is not our goal to achieve state-of-the-art results or to particularly optimize the training process. For this reason, we do not perform hyperparameter tuning or inspect further regularization methods, providing the same base of comparison for all networks. All training details and hyperparameters are available in Appendix A.2 for reproducibility. As future work, we intend to find further evidence of our hypotheses with respect to what is causing the pathologies and study whether we could circumvent them to create more useful deep GNNs.

## 5 Conclusion

This work unwraps the performance stagnation over depth phenomenon (the *flat curve*) which has been verified in the context of current GNN research. We survey and explore both simple and complex/state-of-the-art deep GNNs that empirically show adequate performance and find no evidence that these models are superior to their shallow counterparts. We further justify this phenomenon by showing evidence of pathological behavior in these deep networks, following layer-wise hidden embeddings' inspection, in the form of noisy (Type I pathology) and redundant (Type II pathology) layers. As such, concerning depth in GNNs, the main take home message is that *just because you can, does not mean you should.* For this reason, we encourage other researchers to thoroughly validate the claims that their newly proposed models are able to appropriately scale with respect to depth by measuring the effective gains of these deep models, and propose a depth evaluation protocol to assist them in doing so. Showing evidence of such gains would justify that the pursuit of depth within the field actually has a practical/empirical benefit.

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

## A  Experimental details

### A.1  Benchmarks overview

- Citation networks Yang et al. (2016) are homophilic networks where nodes correspond to scientific papers and links encode citations.

- Coauthor datasets Shchur et al. (2018) are an homophilic graph problem where edges connect authors (nodes) that coauthored a paper.

- Wikipedia datasets Rozemberczki et al. (2021) correspond to heteropilic graphs where nodes correspond to web pages and edges to hyperlinks between them.

- WebKB were introduced in Pei et al. (2020). Similarly to Wikipedia, these networks also encode web pages and hyperlinks.

- The Actor network was also used in Pei et al. (2020) and consists of a sub-problem of the film-director-actor-writer network. In this dataset, an actor-only subgraph is considered, where nodes represent actors and edges define co-occurrences in the same Wikipedia page.

Table 3: Properties of benchmark datasets. Legend: N–Nodes; E–Edges; F–Features; C–Classes; H–Homophily; ED–Edge Density.

| Dataset | # N | # E | # F | # C | H | ED | Type |
|---|---|---|---|---|---|---|---|
| Cora | 2708 | 5429 | 1433 | 7 | .81 | .00074 | Citation |
| CiteSeer | 3327 | 4732 | 3703 | 6 | .74 | .00043 | Citation |
| Pubmed | 19717 | 44338 | 500 | 3 | .80 | .00011 | Citation |
| CS | 18333 | 163788 | 6805 | 15 | .81 | .00049 | Coauthor |
| Physics | 34493 | 495924 | 8415 | 5 | .93 | .00042 | Coauthor |
| Chameleon | 2277 | 36101 | 2325 | 5 | .23 | .0070 | Wikipedia |
| Squirrel | 5201 | 217073 | 2089 | 5 | .22 | .0080 | Wikipedia |
| Actor | 7600 | 33544 | 931 | 5 | .22 | .00058 | Actor |
| Cornell | 183 | 295 | 1703 | 5 | .30 | .0088 | WebKB |
| Texas | 183 | 309 | 1703 | 5 | .11 | .0092 | WebKB |
| Wisconsin | 251 | 499 | 1703 | 5 | .21 | .0079 | WebKB |

We consider the ten dataset partitions of Pei et al. (2020), where each partition consists of randomly splitting the nodes of each class in 60%-20%-20% for training-validation-testing, respectively. An analogous splitting was applied for Coauthor, not considered by Pei et al. (2020).

## A.2 Training conditions

All GNN architectures consist of a single linear layer for input feature transformation into 16 channels, followed by $L$ message-passing layers (or a fixed number of 2 linear layers for the MLP baseline), and a node classification head (linear layer). Besides $L$ and $\alpha$, all hyperparameters are fixed, as Table 4 shows, including the number of hidden channels.

Table 4: Hyperparameters.

| Hyperparameter | Value |
|---|---|
| *Network Architecture* | |
| No. Layers ($L$) | $\{1, 2, 4, 8, 16, 32, 64, 128\}$ |
| Residual strength ($\alpha$) | $\{0.0, 0.1, 0.2, 0.3, 0.4, 0.5\}$ |
| Hidden channels | 16 |
| *Training* | |
| Dropout rate | 0.5 |
| Weight decay | 0.0005 |
| Max. Epochs | 500 |
| Early Stopping (patience) | 50 |
| Learning rate | 0.01 |
| Optimizer | Adam |

This setup means that each $\langle L, \alpha \rangle$ pair defines a different GNN, culminating in a total of 48 architectures. We run this setup for all ten partitions of each dataset as a 10-fold cross-validation process. This results in a total of $48 \times 11 \times 10 = 5280$ experiments, for which we log the following, with respect to the test set: 1) the performance in terms of test accuracy; 2) the hidden embeddings, $\mathbf{H}^l$, of all intermediate node representations after each convolution layer $l \in \{1, ..., L\}$.

# B  Further analyses

## B.1  GCN+InitRes Ablation Study

### B.1.1  Statistical Tests for Performance Comparison

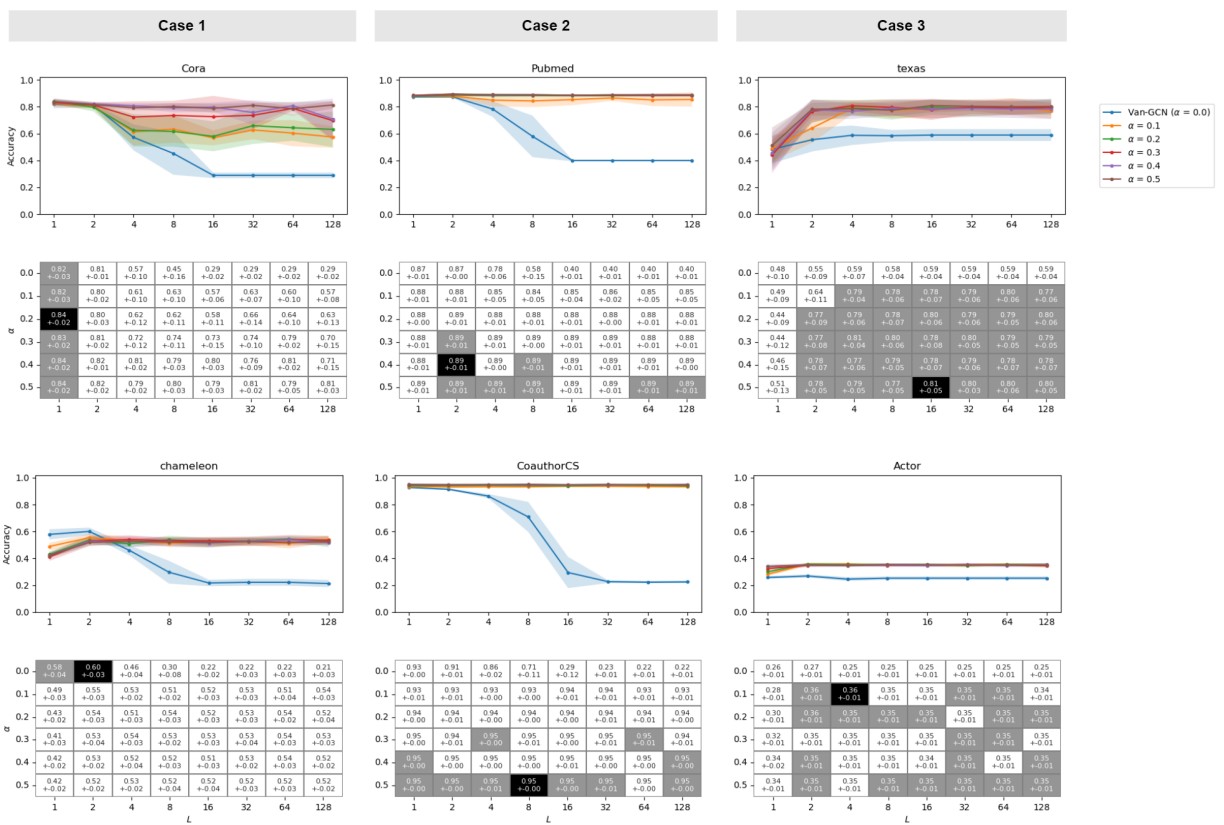

Figure 4: Results of the ablation study of depth ($L$) vs. residual connection strength ($\alpha$) in terms of node classification accuracy for 2 benchmark examples of each case. [**Plots**] Mean test accuracy over depth per studied $\alpha$ (faded areas correspond to standard deviation). [**Performance matrices**] Mean accuracy $\pm$ standard deviation for each $\langle L, \alpha \rangle$ architecture: black shading corresponds to $\mu^d_{\textbf{best}}$; grey shading highlights the cases which satisfy the null hypothesis, $\mu^d_{\langle L, \alpha \rangle} = \mu^d_{\textbf{best}}$.

We compare the average test accuracy of the different network architectures on each benchmark dataset through statistical tests. Let us consider that each dataset $d$ is associated with a set $A^d_{\langle L, \alpha \rangle} = \{acc_1, ..., acc_{10}\}$, where $acc_i$ refers to the test accuracy of the network defined by $\langle L, \alpha \rangle$ on the $i$-th partition of $d$. Let us yet define $\mu^d_{\langle L, \alpha \rangle}$, the average of all elements in $A^d_{\langle L, \alpha \rangle}$. We can finally define $\mu^d_{\textbf{best}}$ which corresponds to the network architecture leading to the largest average test accuracy in $d$, $A^d_{\textbf{best}}$. This set is taken as a reference of the best performance in $d$, against which all other sets $A^d_{\langle L, \alpha \rangle}$ shall be compared. For that purpose, we conduct a paired $t$-test on the following hypotheses:

- $H_0$ (*null hypothesis*): $\mu^d_{\textbf{best}} = \mu^d_{\langle L, \alpha \rangle}$;

- $H_a$ (*alternate hypothesis*): $\mu^d_{\textbf{best}} \neq \mu^d_{\langle L, \alpha \rangle}$

We consider a 95% confidence interval, thus rejecting the null hypothesis for $p$-values $< 0.05$. This means that for $p$-values $> 0.05$ we cannot discard with statistically significant relevance that the averages under comparison are equal, i.e. equivalent to top performance in the cases under analysis.

Figure 4 shows the performance over depth for different values of $\alpha$ in two views: 1) a plot chart of mean accuracy $\pm$ standard deviation over number of layers; 2) a matrix of accuracies per $\langle L, \alpha \rangle$ network, our search space. We conduct paired $t$-tests to find evidence of statistically significant differences of performance between each $\langle L, \alpha \rangle$ and the architecture leading to the best absolute mean accuracy for each dataset. Shaded areas correspond to architectures for which we could not reject the null hypothesis.

The chart view evidences the *flat curve* behavior that we aim to explain. We can further observe a relation between the magnitude of $\alpha$ and the performance drop to a semi-constant accuracy range over depth, more evident for some benchmarks than others (e.g. Cora). This proves that the behavior under analysis is directly tied to our smoothing/sharpening weighting coefficient, which in this case corresponds to controlling the initial residual connection strength. At the same time, we observe equivalent accuracy ranges for all $\alpha$ larger than a certain, dataset-specific threshold, especially for deeper networks.

We can also see different patterns with respect to $\alpha$ for each of the cases outlined in Table 2 by analysing the matrix view. Coherently to what we saw before, Case 1 strongly benefits from the smoothing component, which can even be optimal after a single graph convolution, regardless of the value of $\alpha$, or can only be achieved for $\alpha = 0$. Case 2 benchmarks benefit from high residuals ($\alpha \geq 0.3$); and Case 3 benchmarks achieve top performance for any $\alpha \geq 0.1$. On the other hand, by inspecting the depth dimension, we can observe that shallow versions are mostly preferred in Case 1, but top performing architectures for Cases 2 and 3 are mostly independent of depth.

### B.1.2 Model comparison methodology

The comparison basis for the results of Table 2 was best performance for each model type. We considered the same number of hidden dimensions for all networks, meaning that the number of parameters of each network varies only with depth. For GNNs, we ran a grid search of $\alpha$ and depth and report the results for the best performing ($\alpha$, depth) pair (see per shown in B.1.1). For vanilla-GCN, this meant that Table 2 shows results for 1- or 2-layers networks for all benchmarks, with the exception of Texas (4-layer GCN was reported because the absolute average of accuracy was superior; however, no statistically significant difference was found compared to the 2-layers version). For the MLPs, we considered networks of 2-layers, as per most frequently seen in the literature in benchmarking analysis with these datasets.

### B.2 Description of Deep GNN Layers

Table 5 provides a description of all terms and coefficients of the layers of the surveyed deep GNNs.

GCNII and $G^2$ models were considered in our empirical analyses, and we show evidence of pathologies for some of their deep versions, following the application of our depth evaluation protocol, for exemplification. Despite not having considered GPRGNN and GGCN in these analyses, we have reasons to believe that analogous pathologies can be happening in their case. For example, the ablation studies on $\gamma_l$ published in the original GPRGNN publication Chien et al. (2021) show that this variable frequently tends to zero as depth increases (meaning that the deeper, more smoothed representations will be negligible for attaining the final representations; thus, we should expect hidden embeddings of high similarity as we go deeper, making them redundant–Type II pathology). For the GGCN case, the complexity of the layers can hamper its close analysis; in order to be thorough, it would be important to look into all variables (learned or tuned) and study their downstream impact. However, just by looking into $\eta$, we find that this function easily tends to zero after a small number of iterations (which can be tuned), again making the smoothing term nearly irrelevant as we go deeper. These analyses, however, are mostly hypotheses and should be further investigated through the application of dedicated protocols.

### B.3 Depth Evaluation: further examples

Figures 5, 6 and 7 show the results of our depth evaluation protocol for 16-layers GCN+InitRes, GCNII and $G^2$ for all benchmarks of Cases 1, 2 and 3, respectively. Figure 8 shows an example of the severity of pathological behavior particularly observed for benchmarks of Case 3, for progressively deeper networks. Figure 9 showcases the pathological behavior observed in 16-layers GAT+InitRes networks.

Table 5: Description of the terms and coefficients of the surveyed layers. These layers have been empirically shown to enable deep architectures. Please refer the original publications for further information.

| Method | Layer | |
|---|---|---|
| GCNII (Chen et al., 2020) | $$\mathbf{H}^l = \sigma\Big(\big((1-\alpha^l)\dot{\mathbf{A}}\mathbf{H}^{l-1} + \alpha^l\mathbf{H}^0)((1-\beta^l)\mathbf{I} + \beta^l\mathbf{W}^l\big)\Big)$$ | |
| | $\alpha^l$ | Scalar (hyperparameter) |
| | $\beta^l$ | Scalar (hyperparameter) |
| GPRGNN (Chien et al., 2021) | $$\mathbf{H}^L = \sum_{l=0}^{L} \gamma_l \dot{\mathbf{A}}\mathbf{H}^{l-1}$$ | |
| | $\gamma_l$ | Scalar (learned) |
| GGCN (Yan et al., 2022) | $$\mathbf{H}^l = \mathbf{H}^{l-1} + \eta\bigg(\sigma\Big(\alpha^l(\beta_0^l\hat{\mathbf{H}}^{l-1} + \beta_1^l(\mathbf{S}_{pos}^l \odot \dot{\mathbf{A}} \odot \mathbf{T}^l)\hat{\mathbf{H}}^{l-1} + \beta_2^l(\mathbf{S}_{neg}^l \odot \dot{\mathbf{A}} \odot \mathbf{T}^l)\hat{\mathbf{H}}^{l-1}\Big)\bigg)$$ | |
| | $\eta$ | Decay function (based on hyperparameters) |
| | $\alpha^l$ | Scalar (learned) |
| | $\beta_0^l$ | Scalar (learned) |
| | $\hat{\mathbf{H}}^{l-1}$ | $\mathbf{H}^{l-1}\mathbf{W}^{l-1} + \mathbf{b}^{l-1}$, $\mathbf{b}$ (bias) $\in \mathbb{R}^{n \times m}$ |
| | $\beta_1^l$ | Scalar (learned) |
| | $\mathbf{S}_{pos}^l$ | Positive messages matrix, $\mathbb{R}^{n \times n}$ |
| | $\mathbf{T}^l$ | Degree correction matrix, $\mathbb{R}^{n \times n}$ |
| | $\beta_2^l$ | Scalar (learned) |
| | $\mathbf{S}_{neg}^l$ | Negative messages matrix, $\mathbb{R}^{n \times n}$ |
| G$^2$ (Rusch et al., 2023b) | $$\mathbf{H}^l = (1-\tau^l) \odot \mathbf{H}^{l-1} + \tau^l \odot \sigma\left(\dot{\mathbf{A}}\mathbf{H}^{l-1}\mathbf{W}^l\right)$$ | |
| | $\tau^l$ | Rates matrix (learned), $\mathbb{R}^{n \times m}$ |
| | $l$ | Layer |
| | $n$ | Number of nodes |
| | $m$ | Number of channels |
| | $\dot{\mathbf{A}}$ | Normalized adjacency matrix, $\mathbb{R}^{n \times n}$ |
| | $\mathbf{H}^l$ | Hidden embeddings matrix of the $l$-th layer, $\mathbb{R}^{n \times m}$ |
| | $\mathbf{W}^l$ | Weights matrix of the $l$-th layer, $\mathbb{R}^{m \times m}$ |
| | $\sigma$ | Activation function (e.g., ReLU, softmax) |

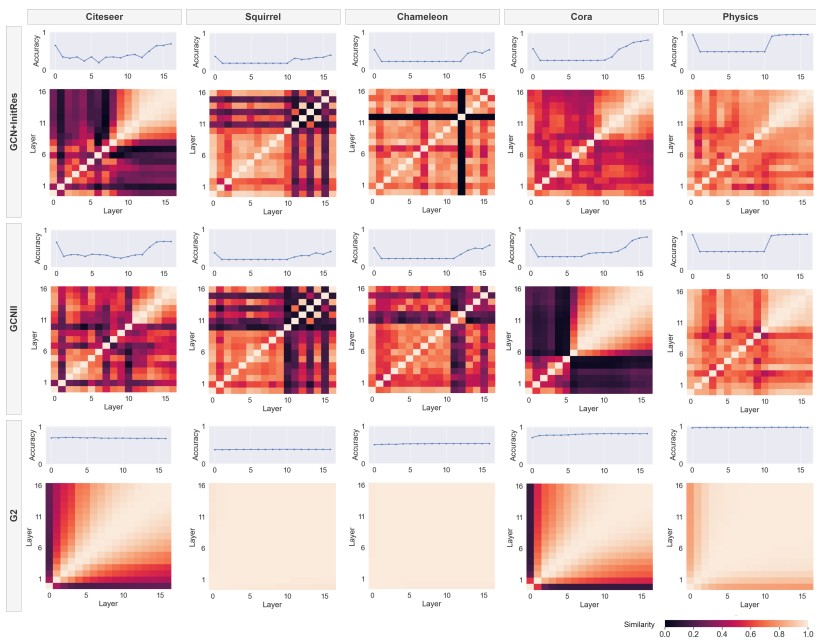

Figure 5: Results of our depth evaluation protocol for GCN+InitRes, GCNII and $G^2$ (16-layers) on Case 1 benchmarks (Citeseer, Squirrel, Chameleon, Cora, Physics).

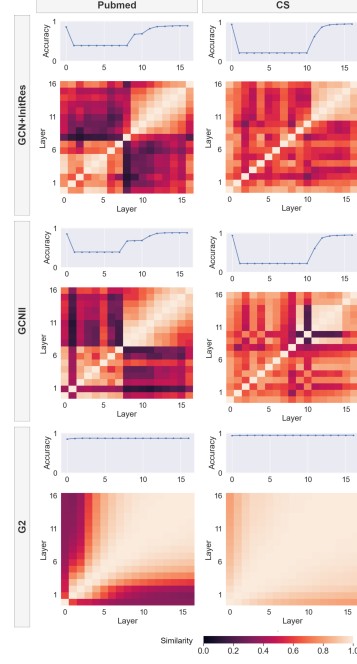

Figure 6: Results of our depth evaluation protocol for GCN+InitRes, GCNII and $G^2$ (16-layers) on Case 2 benchmarks (Pubmed, CS).

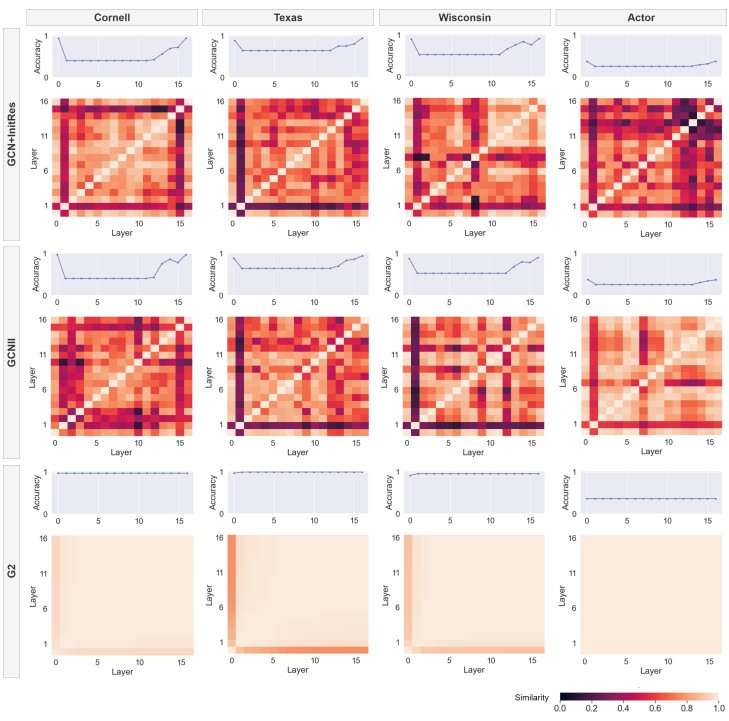

Figure 7: Results of our depth evaluation protocol for GCN+InitRes, GCNII and G$^2$ (16-layers) on Case 3 benchmarks (Cornell, Texas, Wisconsin, Actor).

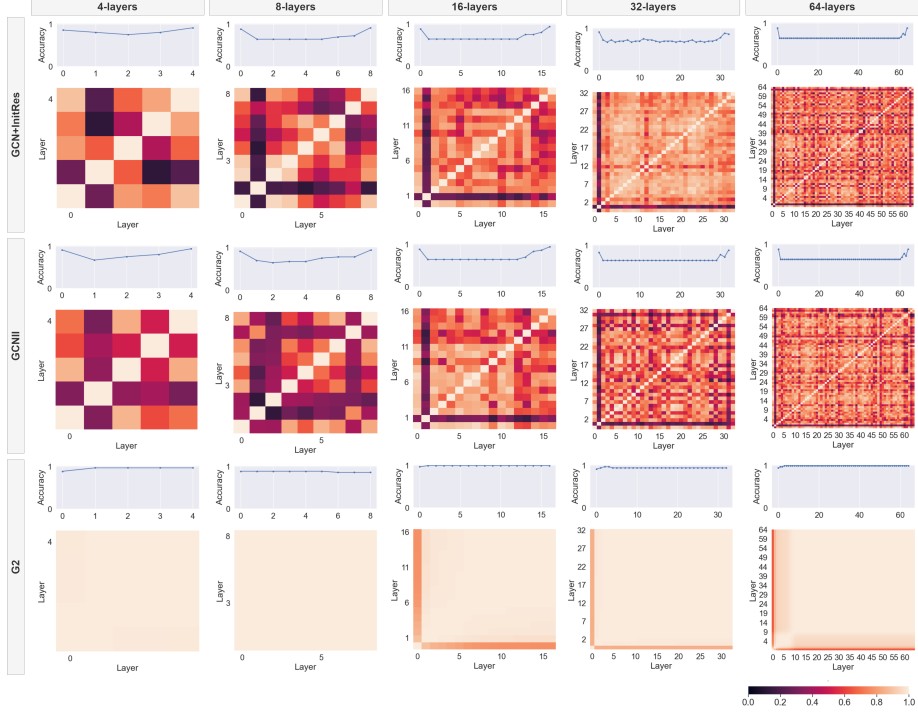

Figure 8: Results of our depth evaluation protocol for GCN+InitRes, GCNII and G$^2$ (4/8/16/32/64-layers) on Texas benchmark, as an example of severe cases of Type I and Type II pathologies for all studied depths.

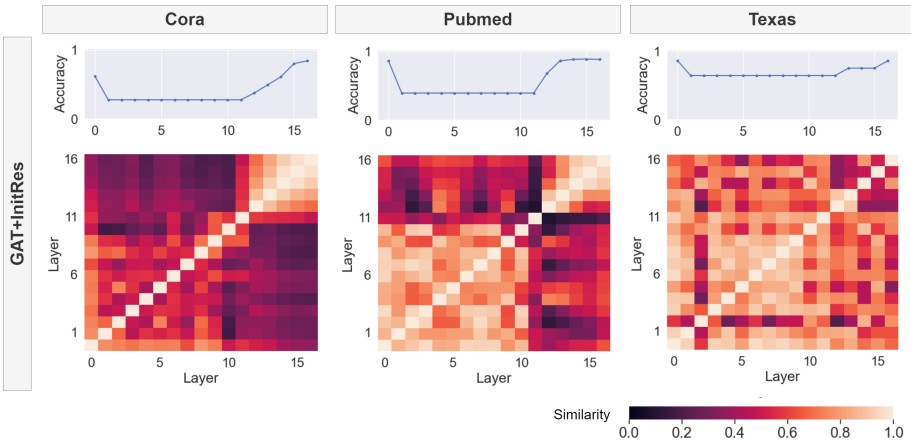

Figure 9: Results of our depth evaluation protocol for GAT+InitRes (16-layers) on Cora, Pubmed and Texas benchmarks. The implemented architecture of GAT+InitRes is analogous to that of GCN+InitRes (as described in the main paper). Similar trends of pathological behaviour are verified.

