# OpenReview forum: "Depth Scaling in Graph Neural Networks: Understanding the Flat Curve Behavior"
_TMLR — Accepted by TMLR_

### Review · Reviewer_2Eab · 2024-02-25

**Summary Of Contributions:**

This work studies the problem that, while current approaches for deep GNNs are able to successfully avoid oversmoothing, their performance (or discriminate power) does not improve with depth, which the authors refer to as a "flat curve" behaviour).

To this end, the authors provide an intuition on what current approaches do, and analyse one of them on a number of datasets to distinguish three types of dataset types. Then, the authors provide a protocol to evaluate the healthy behaviour of deep GNNs, and provide several scenarios where current methods exhibit two pathological behaviours: one where previous layers are ignored, and another where deeper layers are ignored, behaving like identity functions.

Finally, the authors provide a discussion on the possible causes of this behaviour, as well as on how the observed pathologies should generalize to other GNNs, or the role that residual connections have on the matter.

**Audience:**

Yes

**Broader Impact Concerns:**

I do not have any broader impact concerns in this work.

**Claims And Evidence:**

Yes

**Requested Changes:**

- C1. The motivation of the paper needs to be significantly stronger. As of now, I see two main flaws in the premise that need to be justified:
  - A real example: One could argue that in all examples either simple GNNs or MLPs reach sota results on and, hence, it is unreasonable to expect deeper GNNs to make use of the extra layers. The authors should find stronger datasets of the type "Case 2", maybe those from [OGB](https://ogb.stanford.edu/) may work.
  - Additional information: If Type II is a real pathology, I understand the classification performance worsens. If that's the case, plots like those in Fig. 2 should show the final performance. Otherwise, I would argue G2 is a perfectly healthy model.

- C2. A somewhat reasonable hyperparameter tune should be performed.

---

After the rebuttal with the authors, I consider that the additional experiments are just enough to empirically validate the claims of this work.

**Strengths And Weaknesses:**

**Strengths**
- S1. The paper is clearly written, easy to follow, and the methods evaluated properly introduced,

- S2. The protocol to evaluate depth performance is sensible and novel, yet I would add final performance as well (see C1).

- S3. It is quite interesting to observe that Type I pathology happens in certain type of GNNs.

- S4. Of having a stronger motivation/premise, the conclusions of the papers would be quite insightful, encouraging the community to not pursue depth for the sake of it. It even made me consider if a similar behaviour happens in non-GNN residual connections.

**Weaknesses**

- W1. My main concern directly goes to the premise of this work. While I understand that "more layers = better performance" is a desirable goal, I'd argue this is not a general thought by the community (at least it is not my case). If the network is able to classify with 10 layers, it is completely fine if the next 5 layers are identity functions. Put differently, to me "Type II Pathology" looks like a higher-resolution version of "Healthy behavior" in Fig. 2 (as long as classification performance is equal). I believe there are no strong enough arguments to justify the premise of the paper. See requested changes for more details.

- W2. As mentioned in the paper, there is no hyperparameter tuning. While I understand there are lots of experiments, it is hard to take any conclusions of the work as true using the same hyperparameters for 1- and 128-layer GNNs.

- W3. The conclusions regarding residual connections (e.g., those from section 4.3) while interesting in this context, I do not find them as novel. I'd argue that this intuition is built whenever residual connections are first taught, e.g., using the prototypical example of learning an identity with a network, which needs to leverage previous representations.

- W4. While existing approaches to train deep GNNs are correctly explained, there is no discussion regarding previous works on understanding the role of depth on GNNs at all. For example, [Not too little, not too much: a theoretical analysis of graph (over)smoothing](http://arxiv.org/abs/2205.12156) studies theoretically the role of layers in regression problems with linear GNNs, and shows that in that case *there exists* an optimal depth, and that keep adding layers would lead to constant features (which goes into the argument that Type II is actually desirable).

---

> ### Author Response · Authors · 2024-03-06
>
> We would like to thank the reviewer for appreciating the conclusions of our work and for the important points of discussion that were brought forward. (We will split the discussion in 2 comments, due to characters limit.)
>
> **W1.** Thank you for bringing up this important discussion.
>
> We agree that having layers that are identity functions is not a problem in itself. In fact, it can be a desired behaviour, for example for avoiding the need to tune depth during training. However, we believe it is important that the community explicitly realises that this is what is happening (please refer also to the discussion of W3). Several papers – including the ones we surveyed – claim to achieve depth scaling with novel architectures, but it seems that these deeper networks result in either noisy or identity layers – what we call pathological behaviour. Our claim here is that we are not being successful in depth scaling if we do not verify a corresponding increase in performance (or, at least, some concrete evidence of it for specific cases), which suggests that these novel architectures offer little advantage over their analogous shallow versions.
>
> Going deeper into the categorization of this behaviour as pathological, we must highlight that, while we did not find a Type I pathology-like behaviour documented anywhere else, what we classified as a Type II pathology is a behaviour that has been previously reported (and identified as a motive for pathological depth) for CNNs in the original CKA paper. In this context, the authors found that plain-CNNs showed evidence of this behaviour, while res-CNNs did not [Kornblith et al., 2019]. For that reason, we cannot take credit for being the first to define such behaviour as pathological.
>
> It is not clear in [Kornblith et al., 2019] if the authors particularly considered the (modest) performance degradation as part of the definition of such pathology or not – the plateau in accuracy and the high similarity of hidden embeddings are the main discussion points. However, in our understanding, even if performance does not decrease but remains equivalent, this behaviour is, at the very least, an inefficiency, wasting valuable resources that include computation time and allocated memory. It can also lead to unproductive searches for the optimal depth in architecture tuning, as it seems that optimal smoothing can still be achieved with a small number of layers. Making researchers aware of this phenomenon could potentially save valuable development time by encouraging them to create more informed and narrower search spaces. Therefore, we characterize this behaviour as pathological when the goal is explicitly to achieve better performance through increased depth.
>
> We believe that it is of utmost importance that researchers of the field are aware of the identified phenomena for GNNs in particular. We expressed the main motivation for our work in the 3rd paragraph of the introduction. The fact that deeper networks are resorting to random or identity functions in most of their layers to reach an optimal level of smoothing that could be achieved with a significantly smaller number of layers also questions the current motivation for pursuing depth as a means to capture useful long-range dependencies of information k-hops away. It is difficult to maintain that the community is being successful in such task if we do not find evidence of improved expressivity in deep networks.
>
> In summary, the premise of this work is precisely that the community’s efforts of pursuing depth as a means for better expressivity of the network is not delivering this desired outcome, and we should not pursue depth for the sake of it (a message that we seem to have been successful at conveying, as per the comment in S4). We reiterate that, despite the efforts to scale GNNs with respect to depth, we are not improving expressivity; we show empirical proof of the equivalence of shallow nets to the corresponding deep versions (either by creating noisy layers or redundant layers/identity functions). We intend to encourage future works claiming depth scaling in GNNs to quantify the effective gains in terms of classification performance with thorough analyses, as proof of such gains would bring much desired novelty to the field.
>
>
> **W2.** We prioritized the inspection of network architecture parameters in our experiments (depth, α). However, we understand that the extent of our conclusions can be questioned without some further tuning of the training hyperparameters. For that reason, we are currently running a search over a relevant hyperparameter space (for all studied depths). We shall then update the results wherever appropriate (in particular, the results of Table 2). Following some of our own manual tuning process, we do not anticipate significant changes in the reported results and, more importantly, in the main conclusions of the work.

---

> ### Author Response · Authors · 2024-03-06
>
> **W3.** The advantages of residual connections, namely how these facilitate training and their role in enabling depth scaling, are indeed well-known and intuitive. Residuals are the identity by default and facilitate training by forcing the network to learn only the deviation from the identity function. We argue that the effective utility of this approach lies in leveraging them to create more powerful and discriminative models. There is little practical benefit in learning identity functions alone to solve a classification task, which is the behaviour we verify for some GNNs.
>
> Moreover, depending on the type of base network architecture used with residuals, we seem to be able to identify different behaviours following hidden embedding inspection. In fact, [Kornblith et al., 2019] verified that plain-CNNs manifested identity layers, while res-CNNs did not. Our work, on the other hand, shows that some types of res-GNNs show such behaviour. Given this context, we argue that the impact of residuals in learning is not obvious or even transversal to all types of neural architectures, and we can only get a better grasp of what is effectively happening by inspecting the hidden embeddings. For example, in our understanding, the role of residuals in CNNs leads to a more desired outcome as compared to residuals in GNNs: res-CNNs progressively learn better representations while res-GNNs create noisy or identity layers (enabling them to find optimal smoothing without tuning depth, but not adding to their discriminative power). We believe this is an important message for the community, as researchers may expect that what was true for CNNs can also hold for GNNs.
>
> In section 4.3 in specific, we claim novelty in the disclosure of the pathologies as a justification for a well-known problem in the literature that had not been fully explained before – how some heterophilic graphs pose challenges for GNNs, which often perform on-par with MLPs in such cases. By inspecting how deep networks learn (as the motivation for our experiments), we discovered that the same phenomena that explain inefficient depth scaling can also explain the equivalence to MLPs in this case: if neighbourhood aggregation through graph convolutions does not lead to improved representations for the downstream task, networks will learn to resort to noisy or identity layers (i.e., in this case, optimal smoothing is found for #layers = 0). While this seems evident following this work’s introduction of the pathologies, it is a problem that had not been fully explained in the literature before, to the best of our knowledge, which is why we claim that providing a justification for this phenomenon is a novel contribution.  We will further clarify this in the manuscript
>
> **W4.** Thank you for pointing out that this discussion could be valuable. There are indeed some previous works on the theoretical role of depth that inspired us to look for the empirical evidence we introduce in this work. Poor GNN performance due to apparent oversmoothing or loss of expressivity in deeper GNNs has been investigated in some works, while other authors argued that it is not oversmoothing, but rather the training difficulty of GNNs that leads to poor results. This adds to the argument that the way GNNs learn is still poorly understood, often leading to contradictory information following theoretical reasonings.
>
> Our work brings complementary insights from an empirical perspective, which can help to interpret the theoretical contributions of previous works. As the reviewer mentioned, combining the insights of our work with those of the literature, we can for example understand that Type II can be desirable, but mostly in case one wants to avoid fine-tuning depth. However, researchers should be aware that this is the main reason for adding residuals to deep GNNs; the performance increase expected for deeper res-CNNs, for example, is not verified for deep res-GNNs.
>
> We will add some of these references to the manuscript where appropriate.

---

> ### Author Response · Authors · 2024-03-06
>
> **C1. (1. A real example)** Given the premise of our work, which hopefully we were able to clarify with our comment to W1, we mostly opted for the benchmarks used in the works of Table 1 (the original GGCN, GPRGNN, GCNII and G2 papers) to support their claims of depth scaling. These benchmarks consist of 3 orders of magnitude in #nodes; despite that, no change in behaviour occurred. Graph size is not expected to have a big impact in the phenomenon we disclosed. For the sake of this discussion, we ran our protocol for a 16-layer GCNII on a large-scale dataset (ogbn-arxiv) and see evidence of the same behaviour (unfortunately, OpenReview does not allow adding the image to the comment). Other graph properties, however, namely edge density and homophily, can be associated with specific pathology trends, but mainly regarding how many layers of the network will be deemed noisy/redundant (the 3 cases highlighted in the paper are examples of this diversity). We also took precautions to avoid overfitting – early stopping; cross-validation; avoid overparameterization –, as smaller graphs might tend to overfit more easily; we expect that the impact of this effect in our conclusions is very limited.
>
> **(2. Additional information)** Please refer to the clarification of W1, which we hope can address your concerns with respect to the definition of a pathology, particularly those of Type II.
>
>
> **C2.** Please refer to the clarification of W2.
>
>
> Thank you for the valuable input. Please let us know if we addressed your concerns.

---

> > ### Comment · Reviewer_2Eab · 2024-03-10
> >
> > Thank you for the detailed response.
> >
> > Now, for the sake of keeping the discussion short, I will focus on my main concern: the lack of a specific example on which exploiting depth in GNNs gives an edge over shallow GNNs/MLPs (W1). Without a convincing example of this kind, it is hard to make any argumentation regarding type-II pathologies (unless if there is a clear degradation in accuracy, which the authors have still not clarified as far as I can tell).
> >
> > If the performance of type-II pathologies is similar to that of the best-performing methods, the argument to "not pursue depth for the sake of it" becomes weaker. Now, one could argue that it is cheaper to pursue depth on training and prune afterwards, instead of cross-validating the number of layers. While the paper still brings an interesting discussion, the topic (and severity) of it heavily changes.
> >
> > I am thankful to the authors to run extra experiments, but I would appreciate it if they could report (even if in text form) some sort of results to strengthen their arguments: either with a new dataset that shows that depth is necessary; or with an example with type-II pathologies degrading performance with the number of layers.
> >
> > As a sidenote, I do believe that the authors of the CKA paper described type-II as a pathology due to the degradation in performance. Whether it is modest or not (it is 2%, similar to the performance change in Table 2 of the manuscript in review) is something that can vary from reader to reader.

---

> > > ### Author Response · Authors · 2024-03-12
> > >
> > > Thank you for your input and the contributions to this discussion.
> > >
> > > First, we want to clarify that our premise is not that we should see benefits of depth (at least not in these datasets), it is that we find no evidence of these benefits, despite the claims in the original papers. We will further argue this point later in this reply.
> > >
> > > However, for the sake of this argument, and to directly address the reviewer’s concerns, we ran a few performance evaluation analyses on G2 networks, the ones that in our experiments are particularly affected by type-II pathologies. In the original G2 paper, the authors solely relied on Cora benchmark to base their claims of depth scaling (see Figure 3 + “G2 for very deep GNNs” paragraph in section 4 of that manuscript). Let us use the same benchmark and take the performance evaluation methodology a bit further. The following table shows mean and standard deviation accuracy on 10 train-validation-test partitions of this dataset. We followed an analogous methodology to the one we took to compute the results of Table 2 in our paper; we ran statistical analyses testing the null hypothesis that mean accuracy for each depth equals best mean accuracy, rejecting the null hypothesis for p-values < 0.05.
> > >
> > >
> > > ***
> > > **Cora**
> > >
> > > **Depth** ........... **Accuracy (mean±std)** ..... **P-value** ....... **Final assessment**
> > >
> > > L = 1 ...................... 0.788 ± 0.025 ............... 0.000 .............. ≠ Best (worse)
> > >
> > > L = 2 ...................... 0.818 ± 0.019 ............... 0.094 .............. = Best
> > >
> > > L = 4 ...................... 0.829 ± 0.016 ............... 0.761 .............. = Best
> > >
> > > L = 8 ...................... 0.825 ± 0.025 ............... 0.509 .............. = Best
> > >
> > > L = 16 .................... 0.831 ± 0.012 ............... 1.000 .............. Best
> > >
> > > L = 32 .................... 0.812 ± 0.024 ............... 0.040 .............. ≠ Best (worse)
> > >
> > > L = 64 .................... 0.824 ± 0.023 ............... 0.407 .............. = Best
> > >
> > > L = 128 .................. 0.812 ± 0.021 ............... 0.028 .............. ≠ Best (worse)
> > >
> > > ***
> > >
> > > From this table, it is possible to see: 1) performance equivalence for #layers between 2-16; 2) both performance degradation and performance equivalence for #layers > 16. This proves that _type-II pathologies can cause both performance degradation and stagnation if we keep stacking layers_, which, as the reviewer kindly pointed, adds to the severity of this problem. It also adds to the argument that recent works have been claiming successful depth scaling with insufficient benchmarking analyses. The original G2 paper claims, quoting, a “small but noticeable increase in performance for increasing number of layers”; but such claim is based on a single run (or maybe simple average of runs, it is not clear) on a single dataset. Following our analysis, we can find no statistically relevant evidence of such a performance increase on that same benchmark.
> > >
> > > As a final remark, we would like to clarify that we do not advocate that exploiting depth in GNNs gives an edge over shallow GNNs/MLPs. On the contrary, we believe our work contributes to the body of evidence that suggests that significantly scaling depth is not the way to go for GNNs in their current form. Currently, we can find conflicting information regarding the role of depth from theoretical analysis (see, for example, [1], which defends an “optimal smoothing”/depth, vs. [2], which defends there are provable benefits if we keep going deeper). Our work can assist the interpretation of previous theoretical contributions of the literature from an empirical perspective and influence future research directions for addressing a problem that is not trivial and, on occasion, not even generally acknowledged. This context also justifies why proposing a method to improve the benefit of GNN depth scaling or even a method to “alleviate” the potential damaging impact of the pathologies is beyond our goal; however, we do plan to look into the problem of creating more expressive GNNs from a more fundamental point of view as future work.
> > >
> > > We hope this clarification, along with the additional results, can address your concerns. Thank you for the important feedback.
> > >
> > > [1] N. Keriven, “Not too little, not too much: a theoretical analysis of graph (over) smoothing”. Advances in Neural Information Processing Systems, 2022, vol. 35, p. 2268-2281.
> > >
> > > [2] W. Cong, M. Ramezani, and M. Mahdavi, “On provable benefits of depth in training graph convolutional networks,” Advances in Neural Information Processing Systems, vol. 34, pp. 9936–9949, 2021.

---

### Review · Reviewer_Zimx · 2024-03-03

**Summary Of Contributions:**

This paper empirically investigates the existing approaches to tackle the over-smoothing problems in GNNs. While there indeed are various approaches for overcoming over-smoothing problems and enabling the depth-scaling, there is an empirical consensus saying that adding those depths does not lead to performance gain, unlike the other deep neural nets that usually benefit from depth-scaling. The paper provides a thorough empirical evaluation of GCN variants known to tackle over-smoothing problems for several benchmark datasets and proposes a protocol to evaluate the benefits of adding depths to GNNs. The results show that at least judging from the benchmarks tested in the paper, the added depths do not seem to bring significant gain in terms of representational diversity. Actually, even when the GNNs are trained with larger depths, they eventually seem to use only part of the layers or many layers end up being redundant.

**Audience:**

Yes

**Broader Impact Concerns:**

I don't see any ethical concerns or implications may be brought by this work.

**Claims And Evidence:**

Yes

**Requested Changes:**

I must admit that I'm not keen on the state-of-the-art models in GNN literature, but as far as I know, GCN is quite a basic model; there seem to be numerous GNN models, for instance, involving attention layers (GAT) or even transformer based architectures. Would a similar argument also hold for those models? I think the message in the paper can be strengthened by at least mentioning other GNN architectures with different design principles and analyzing them.

**Strengths And Weaknesses:**

Strengths
- The paper highlights an important issue actively discussed in the literature.
- The empirical analysis is extensive and conclusive.
- The analysis based on logistic regression and CKA is interesting, and the justification for type I and II pathologies makes sense.

Weaknesses
- I don't think this should be a main criterion for evaluating this type of paper, but I don't see any technical contribution.
- The main message, "Even if you may add depth, that would not necessarily bring gain", is somehow expected, so the empirical findings are not super surprising.
- While this may be beyond the scope of the paper (but at the same time that is the reason why I think this is a weakness), no insights for designing better GNN architecture for alleviating type I or II pathologies.

---

> ### Author Response · Authors · 2024-03-08
>
> We would like to thank the reviewer for appreciating our analyses and the relevance of our conclusions to the field of GNN research. We hope the next discussion can address your concerns.
>
> **Weaknesses**
>
> **1.** Even though it is not one of the main criterions of evaluation, we would still argue that characterizing the empirical learning behaviour of deep GNNs is a technical contribution, as currently GNNs are still poorly understood. It is even possible to find conflicting information regarding the role of depth from theoretical analysis (see, for example, [1] vs. [2]). Our goal is indeed not to propose a novel architecture, but rather help researchers understand what is happening with current models and influence future research directions for addressing a problem that is not trivial and, on occasion, not even generally acknowledged . Our work can assist the interpretation of previous theoretical contributions of the literature from an empirical perspective.
>
> [1] N. Keriven, “Not too little, not too much: a theoretical analysis of graph (over) smoothing”. Advances in Neural Information Processing Systems, 2022, vol. 35, p. 2268-2281.
> [2] W. Cong, M. Ramezani, and M. Mahdavi, “On provable benefits of depth in training graph convolutional networks,” Advances in Neural Information Processing Systems, vol. 34, pp. 9936–9949, 2021.
>
> **2.** We agree with the reviewer that one should not blindly expect that increasing depth will necessarily reflect in improved performance. However, this has been a trend for other network architectures, namely for res-CNNs, and researchers may expect that what was true for CNNs can also hold for GNNs. There is also an expectation that GNNs’ depth maps directly to the aggregation of information k-hops away, where k = #layers. The fact that deeper networks are resorting to random or identity functions in most of their layers to reach an optimal level of smoothing that could be achieved with a significantly smaller number of layers also questions the current motivation for pursuing depth to this end.
>
> The premise of this work is precisely that the community’s efforts of pursuing depth as a means for better expressivity of the network are not delivering this desired outcome, and we should not pursue depth for the sake of it. Depth scaling is still an active research direction in the field of GNNs (as the works surveyed and analysed in our paper show), which is why we believe that the empirical analyses we disclosed are currently very relevant. We intend to encourage future works claiming depth scaling in GNNs to quantify the effective gains in terms of classification performance with thorough analyses, as proof of such gains would bring much desired novelty to the field.
>
> **3.** This point is indeed beyond the scope of our paper, but we plan to address it in future works (as discussed in the manuscript). However, given the fast-paced advent of novel GNN architectures in recent years, we believe it is important to give researchers the tools to identify where current architectures are failing as fast as possible before we can come up with improvements.
>
> Our intuition is that the problem disclosed in this paper is fundamentally grounded in the message-passing (MP) paradigm and the definition of graph convolutions as smoothing operators in nature. From our experiments, we can see that the smoothing effect of one convolution alone can lead to the best performance on a certain task; this means that we have scenarios of best smoothing for #layers=1. The fact that we go from no graph smoothing (#layers=0) to best smoothing in one single step makes us question whether we are oversimplifying aggregation operations in the spatial domain. Up to this point, the fact that deep GNNs learn the pathologies disclosed in this work has been somewhat useful – it enables them to keep the best smoothing for indefinite depth, preventing performance degradation. But we question if this “best smoothing” is indeed the optimal smoothing and hypothesize that a finer control of this property could lead to more expressive networks. This would mean that the community’s effort should not be on more convolutions (i.e., more depth) but on improving the aggregation method itself, which is why simply “alleviating” the occurrence of the pathologies would bring little gain.
>
> As such, only rethinking MP or finding a new paradigm for aggregation could provide a solution to this problem. This would be a significant shift in this research area and would require significantly more effort than can be achieved in a single paper. Since most efforts in the field of GNN research seem to use this MP/graph convolution paradigm and have been pursuing depth with the expectation of better expressivity, we believe that the message of our work can influence future research directions of the field w.r.t. the fundamentals of GNNs.

---

> ### Author Response · Authors · 2024-03-08
>
> **Requested Changes**
>
> **1.** We opted for analysing the architectures of works that claim depth scaling in our manuscript in the interest of conciseness. However, it is true that by combining the “key ingredients” for depth scaling that we bring forward in section 2.2.3 it is possible to create, for example, deep GAT networks – i.e., provided we add residual connections and smoothing controlling coefficients to each layer (otherwise, oversmoothing still occurs and we will have performance degradation with increased depth). As such, if we implement, for example, a GAT+InitRes architecture (analogous to the GCN+InitRes showed in our paper) we still observe the same behaviour. For the sake of this point, we ran our protocol on a 16-layer GAT+InitRes on Cora, Pubmed and Texas (unfortunately, OpenReview does not allow adding a figure to this comment). The observed behaviour is analogous to what we see in Figure 3 for GCN+InitRes in our manuscript. This goes into the argument that the pathologies are rooted in the “ingredients” for depth scaling, not in the convolution operation itself (which can be more or less expressive in some cases), and goes precisely into our point that we should not resort to methods that enable us to go deep just for the sake of it. We will add this figure in the Appendix, along with these lines of discussion.
>
> Transformers, on the other hand, do not suffer from the same handicaps as the ones discussed in this work, as depth is not considered in the same way, which is why our work is addressed specifically to the GNN research community.
>
>
> Thank you for the valuable input. Please let us know if we addressed your concerns.

---

### Review · Reviewer_Qr7P · 2024-03-04

**Summary Of Contributions:**

The paper investigates why deep Graph Neural Networks (GNNs) fail to significantly outperform their shallower counterparts, a phenomenon termed as the "flat curve" of performance over depth. It critically examines the effectiveness of current depth scaling methods in GNNs and identifies pathological behaviors that may mask rather than solve the underlying issues of depth scaling. Through empirical studies using common benchmark datasets and a case study on GCN with initial residual connections (GCN+InitRes), the research highlights the limitations of deep GNNs in enhancing discriminative power and proposes a depth evaluation protocol to identify and categorize inefficiencies in learning behaviors. The findings suggest that deep GNNs often exhibit redundant or noisy layer behaviors, questioning the pursuit of depth in GNN architectures and urging for more thorough validation of depth scaling claims.

**Audience:**

Yes

**Broader Impact Concerns:**

No ethical justification is needed.

**Claims And Evidence:**

Yes

**Requested Changes:**

As discussed on the weakness and questions above, I would appreciate the authors making changes including:
1. Any potential ideas on improving the GNN depth scaling based on the empirical observations.
2. A better organized Section 4.
3. Clarification on the number of parameters of models compared in Table 2.

**Strengths And Weaknesses:**

**Strength:**
1. This paper included systematic experiments on oversmoothing and depth evaluations.
2. This paper comprehensively discussed the impact of different depth-scaling methods in GNNs. This benchmark is important to show how much concrete progress the GNN community is making toward the real benefits of increasing the GNN depth.

**Weakness:**
1. Although provided with plentiful empirical evidence, this paper did not propose any method to improve the benefit of GNN depth scaling.
2. Section 4 is poorly organized, less motivated, and did not provide any experimental results. I could not get the key message of Section 4 quickly (e.g. why do you want to compare different GNNs).

**Question:**
In Table 2: Do GCN and MLP share a similar number of model parameters for fair comparisons?

---

> ### Author Response · Authors · 2024-03-10
>
> We would like to thank the reviewer for appreciating the relevance of our work in contributing to the progress of GNN research. We hope the next discussion can address your concerns.
>
>
> **1.** Thank you for bringing up this discussion. This comment is closely related to what was pointed out by another reviewer. In order not to paraphrase ourselves, we’d like to refer the reviewer to point 3 of [this comment](https://openreview.net/forum?id=fdyHzoGT8g&noteId=XpyfU9Trjd).
>
> To the direct point of the reviewer’s concern, we would like to clarify that our work goes into the argument that depth is not the way to go for GNNs in their current form. Currently, we can find conflicting information regarding the role of depth from theoretical analysis (see, for example, [1] vs. [2]). Our work can assist the interpretation of previous theoretical contributions of the literature from an empirical perspective and influence future research directions for addressing a problem that is not trivial and, on occasion, not even generally acknowledged.
>
> This context justifies why proposing a method to improve the benefit of GNN depth scaling is beyond our goal, but we do plan to look into the problem of creating more expressive networks from a more fundamental point of view (as discussed in the manuscript). However, given the fast-paced advent of novel GNN architectures in recent years, we believe it is important to give researchers the tools to identify where current architectures are failing as fast as possible so we can collectively come up with improvements that bring significant novelty to the field.
>
>
> [1] N. Keriven, “Not too little, not too much: a theoretical analysis of graph (over) smoothing”. Advances in Neural Information Processing Systems, 2022, vol. 35, p. 2268-2281.
>
> [2] W. Cong, M. Ramezani, and M. Mahdavi, “On provable benefits of depth in training graph convolutional networks,” Advances in Neural Information Processing Systems, vol. 34, pp. 9936–9949, 2021.
>
>
> **2.** Until this point in the manuscript, we showed the equivalence between shallow and deep GNNs and the occurrence of pathologies in these networks. Section 4 aims to further bridge two key points of our work:
> -	Pathologies are a consequence of the addition of residual connections and the smoothing refraining coefficients (as discussed in 3.3.2);
> -	All surveyed methods in Table 1 (i.e., works that claim depth scaling) rely on both of these “ingredients” to create deep architectures (as discussed in 2.2.3).
>
> Because of the direct correspondence between the causes of the pathologies and the identified key ingredients of depth scaling, we expect to see similar behaviour whenever these components are part of deep GNN architectures. We show evidence of this crucial point for the 3 types of GNNs compared in Figure 3, but we believe it is critical that the community understands that our conclusions are not limited to the examples we inspected. For this reason, we expand on the generic relation between the “flat curve” and the pathologies (4.1), on the role of residual connections – one of the most frequently adopted and unquestioned methods in the literature – in GNN learning (4.2), and on how pathologies can not only justify the “flat curve” but also other puzzling phenomena whose causes still lack general agreement in the community (like the equivalence between GNNs and MLPs on some heterophilic benchmarks).
>
> We will clarify this with an introduction paragraph to section 4.
>
> **3.** The comparison basis for the results of Table 2 was best performance for each model type. We considered the same number of hidden dimensions for all networks, meaning that the number of parameters of each network varies only with depth. For GNNs, we ran a grid search of α and depth and report the results for the best performing (α, depth) pair (see detailed methodology and results of the ablation study in Appendix B.1). For vanilla-GCN, this meant that Table 2 shows results for 1- or 2-layers networks for all benchmarks, with the exception of Texas (4-layer GCN was reported because the absolute average of accuracy was superior; however, no statistically significant difference was found compared to the 2-layers version). For the MLPs, we considered networks of 2-layers, as per most frequently seen in the literature in benchmarking analysis with these datasets. As such, we consider this comparison overall fair. We will add this clarification in the Appendix and refer the reader to it under section 2.3-Methodology.
>
>
> Thank you for the valuable input. Please let us know if we addressed your concerns.

---

### Decision · Action_Editor_937A · 2024-04-18

**Recommendation:** Accept as is

**Comment:**

A reviewer noted the lack of datasets and tasks that clearly benefit from deeper GNN architectures. Developing such datasets would be an interesting research direction and valuable contribution to the field.

**Audience:**

Yes, I think the empirical findings (two kinds of pathological behaviors in deep GNNs: redundant or noisy layer behaviors) are interesting and insightful. They could potentially generate new research ideas.

**Claims And Evidence:**

The paper empirically investigates why GNNs do not benefit from having more layers and identifies two pathological behaviors: redundant or noisy layer behaviors. Reviewers find it well-written, easy to follow, and the empirical analysis is extensive, systematic, and conclusive. The issue (questioning the usefulness of depth scaling in GNNs) highlighted in the paper is important, and all reviewers are leaning towards accepting the paper. I agree with their recommendation and vote to accept the paper.